# NK Cell Plasticity in Cancer

**DOI:** 10.3390/jcm8091492

**Published:** 2019-09-19

**Authors:** Sizhe Liu, Payal Dhar, Jennifer D. Wu

**Affiliations:** 1Department of Urology, Feinberg School of Medicine, Northwestern University, Chicago, IL 60611, USA; SizheLiu2022@u.northwestern.edu (S.L.); PayalDhar2019@u.northwestern.edu (P.D.); 2Robert Lurie Comprehensive Cancer Center, Chicago, IL 60611, USA

**Keywords:** NK cells, cancer, plasticity

## Abstract

Natural killer (NK) cells are critical immune components in controlling tumor growth and dissemination. Given their innate capacity to eliminate tumor cells without prior sensitization, NK-based therapies for cancer are actively pursued pre-clinically and clinically. However, recent data suggest that tumors could induce functional alterations in NK cells, polarizing them to tumor-promoting phenotypes. The potential functional plasticity of NK cells in the context of tumors could lead to undesirable outcomes of NK-cell based therapies. In this review, we will summarize to-date evidence of tumor-associated NK cell plasticity and provide our insights for future investigations and therapy development.

## 1. Introduction 

Natural killer (NK) cells play important roles in anti-tumor immunosurveillance. Using an array of germline-encoded activating and inhibitory receptors, NK cells are able to detect and kill malignant cells without prior sensitization by sensing the altered surface ligand repertoire on tumor cells [1]. NK cells recognize tumor cells via two major mechanisms [1,2]. One is through the “missing-self” mechanism, whereby NK cells recognize tumor cells that have downregulated self major histocompatibility complex I (MHC I) molecule. In the healthy physiological state, NK cells express an array of inhibitory receptors, such as killer cell inhibitory receptors (KIRs) and CD94/NKG2A complex, that interact with major histocompatibility complex I (MHC I) molecule to deliver an inhibitory signal to NK cells and sustain NK cells in a quiescent state. Tumor cells often lose or have downregulated MHC I expression, which in turn diminishes the inhibitory signals in NK cells and leads to NK cell activation. The second is through the “induced-self” mechanism, whereby NK cell activating receptors engage their ligands “induced” or overexpressed on tumor cells. Ligands for NK cell activating receptors, such as NKG2D, natural cytotoxicity receptors (NCRs), and DNAM-1 (DNAX accessory molecule), are generally absent in normal tissues, but induced on tumor cells in response to various oncogenic stress [1,2]. In the context of therapeutic antibodies, NK cells can also recognize tumor cells through engaging CD16, the receptor for Fc region of certain isotype of antibodies, to mediate antibody-dependent cellular cytotoxicity (ADCC) against tumor cells [1]. NK cells are also an important component in shaping adaptive immune responses through secreting an array of cytokines, such as IFN-γ, TNF-a, GM-CSF, and interleukin (IL)-10, and interacting with dendritic cells [3,4,5,6,7].

A number of clinical studies have demonstrated the significance of NK cells in controlling cancer. A large-scale, 11-year follow up study showed that higher NK activity correlates with lower cancer risk [8]. Early clinical trials of NK adoptive transfer in acute myeloid leukemia (AML) patients have presented a beneficial, although limited role in controlling the disease [9,10]. In the recent IMMUNOSTIM study (NCT00478985), a significantly higher count of CD56^dim^ cytotoxic NK cells was found in non-relapsing as opposed to relapsing chronic myeloid leukemia (CML) patients after imatinib (a tyrosine kinase inhibitor, TKI) discontinuation [11]. In two other independent multi-center clinical trials (DADI trial and EURO-SKI, NCT01596114) in CML patients, a higher rate of disease remission after TKI discontinuation was associated with a higher percentage of activated NK cells after TKI discontinuation [12,13]. In patients with colorectal cancer, gastric cancer, and lung cancer, NK infiltration is associated with better outcomes [14,15,16,17]. High NK cell activity also correlated with better clinical response with increased progression-free survival (PFS) to trastuzumab in metastatic breast cancer patients and delayed progression to castration-resistant diseases in prostate cancer patients [18,19]. A gene signature correlating with increased NK infiltration was associated with improved survival in melanoma patients [6,20]. These studies support the significance of NK cells in controlling cancer. NK cells are thus considered an attractive target for immunotherapies. However, a highly immune suppressive tumor microenvironment can alter NK cell phenotype and function and result in their inability to effectively exert their anti-tumor activity. Moreover, a number of recent studies have presented the potential pro-tumorigenic activities of NK cells, owing to their functional plasticity in the context tumor. With many ongoing endeavors in developing NK cell-based therapies clinically and preclinically, it is critical to understand these various aspects of NK cells. In the following sections, we will summarize current clinical efforts for NK cell-based therapies and recent evidence of tumor-induced NK cell plasticity, with a focus on tumor microenvironment (TME)-induced NK cell phenotypical and functional alternations.

## 2. Current NK Cell-Based Pre-Clinical and Clinical Studies

NK cell-based therapies have been evaluated in clinical trials [21,22,23]. To date, two main types of NK-based therapies have been pursued: (i) adoptive transfer of NK-92 cell line and cord blood derived-NK cells expanded and activated ex vivo; (ii) use of engineered chimeric antigen receptor (CAR)-NK cells. Immune-stimulatory cytokines, such as IL-2 and IL-15, have been used in the therapeutic settings to potentiate NK cell survival and function. NK cell-based therapies generally presented good safety profiles in clinic. How to achieve desired therapeutic efficacy of NK-cell based therapies is still a challenge. Thus far, the most effective indications for NK cell-based therapies are limited to hematological malignancies. Both pre-clinical and clinical studies have shown that NK cell-based therapies can promote graft-versus-leukemia or graft-versus-tumor effect, without provoking graft-versus-host disease (GvHD) [9,24,25,26]. To improve the therapeutic efficacy of NK cell-based therapies, Ruggeri et al. have pioneered allogenic haploidentical NK cell transfer by selecting patients with KIR ligand mismatch to allow alloreactive NK cell activation. This approach presented to be safe and provoked protection against acute myeloid leukemia (AML) relapse [27]. Miller and colleagues further explored this haploidentical NK cell transfer strategy, along with immune suppressive cyclophosphamide and fludarabine regimens in patients with metastatic melanoma, Hodgkin’s disease, renal cell carcinoma, and poor-prognosis AML [9]. The strategy is generally safe and achieved a good in-patient NK cell expansion. However, efficacy was only presented in AML patients after receiving intensive immunosuppressive regimen of cyclophosphamide and fludarabine, but not in patients receiving low intensity immunosuppressive regimens [9]. Five out of 19 acute myeloid leukemia (AML) patients attained complete remission; however, high doses of interleukin-2 (IL-2) and high conditioning regimens triggered severe hematological as well as non-hematological toxicities in the patients [9]. To address such toxicity issues and to evaluate the suitability of low-intensity immunosuppressive regimens, Rubnitz and group conducted a pilot study (NCT00187096) in children with AML. The study concluded that the low-dose immunosuppressive regimens in patients receiving donor-recipient KIR-HLA mismatched NK cells were safe and feasible [24]. These initial studies generated a great interest in pursuing NK cell-based cancer immunotherapy.

The outcome of clinical trials using NK cell-based therapies for solid tumors has been less optimistic than for hematological malignancies. Although phase I clinical trials demonstrated the safety of adoptively transferred NK cells [28,29,30,31,32], clinical efficacy reported to date is limited. A phase II study (NCT00376805) conducted by Miller and group using adoptively transferred allogenic NK cells to treat patients with recurrent ovarian and breast cancer demonstrated limited efficacy [33]. In this study, 20 patients underwent fludarabine and cyclophosphamide regimen with or without low-dose total body irradiation (TBI) before NK cell infusion. Although donor DNA was detectable after seven days of NK cell infusion in 69% of the patients without TBI and 85% of the patients with TBI, the infused cells did not persist beyond seven days. There was an undesirable expansion of immunosuppressive regulatory T cells (Tregs) in response to low dose IL-2 (10 million unit/dose, 3 times per week, 6 doses in total) that was given in conjunction with NK cell infusion [33]. To avoid IL-2 associated Treg expansion, IL-15 is now being explored as an alternative for NK expansion and activation [34,35]. The clinical efficacy of different NK-based therapies in solid tumors remains to be determined in ongoing phase II clinical trials (e.g., NCT03958097, NCT03007823, NCT02409576, NCT03242603). 

The lack of efficacy of primary NK cells was thought in part to be the result of suboptimal expansion of primary NK cells. The development of an optimal protocol to expand primary NK cells ex vivo for therapeutic use remains to be a challenge [36,37,38]. To overcome the challenge, many attempts using the NK cell line, NK-92, or the CD16-engineered NK-92 derivatives, have been made in clinical trials [39,40,41,42,43]. In comparison with primary NK cells, NK-92 can be expanded easily and indefinitely and has been proposed to be an “off-the-shelf” resource for immunotherapy [31,42,44,45]. Despite these benefits, irradiation of NK-92 cells is required prior to infusion into patients in order to prevent the possibility of malignant transformation [46], as the cell line was originated from a non-Hodgkin lymphoma patient [39]. This manipulation may limit the persistence of the infused cells and long-term anti-tumor therapeutic efficacy in patients [47]. 

Chimeric antigen receptors (CARs) are engineered receptors with an extracellular antigen-recognition domain for specific tumor antigens, connected with intracellular co-stimulatory and signaling domains that trigger lymphocyte activation upon antigen binding. CAR engineered NK-92 cells and primary NK cells have been evaluated in preclinical studies both in hematologic malignancies and solid tumors [48]. A number of clinical trials are underway to evaluate the potential of (CAR)-NK92 and CAR-primary NK cell therapies in refractory CD19^+^ malignancies, leukemia, and lymphomas, as well as myeloid malignancies in children and adults [48,49]. To overcome the limitations of using NK-92 cells and the challenge of manipulating primary NK cells, human induced pluripotent stem cell (iPSC)-differentiated NK cells have been used to construct iPSC-NK-CAR as an “off-shelf” CAR-NK strategy and tested in preclinical settings. It was shown that iPSC-CAR-NK therapy was significantly more effective than CAR-T therapy in reducing the tumor burden in mice bearing xenografted ovarian tumors [50]. Further, a reduced systemic toxicity, such as weight loss and cytokine release syndrome, was observed in mice receiving iPSC-CAR-NK treatment as compared with CAR-T treatment. This study presented the proof-of-concept that iPSC-NK-CAR can be a potential off-shelf cancer immunotherapeutic approach. Evidently, clinical evaluation of this therapeutic approach is warranted. 

## 3. NK Cells Are Composed of Phenotypically and Functionally Diverse Subsets

Human NK cells are classically divided into two major subsets, CD56^dim^CD16^hi^ and CD56^bright^CD16^−/lo^ [51,52,53]. NK cells in human peripheral blood are composed of approximately 90% of CD56^dim^CD16^hi^ and 10% of CD56^bright^CD16^−/lo^ [51]. The two subsets of NK cells are thought to differ in maturation status and functions. The CD56^dim^CD16^hi^ subset is considered mature or terminally differentiated NK cells and predominantly exerts cytolytic function. The CD56^bright^ CD16^−/lo^ NK cell subset is considered immature NK cells being less cytolytic, but has more immune regulatory function through cytokine secretion [51,53]. The CD56-based maturation and functional characteristics of human NK cells were supported by recent characterization based on the surface density of CD27 and CD11b, similar to the maturation characterization of mouse NK cells [54,55]. The increasing levels of maturity of human NK cells were characterized as CD11b^−^CD27^−^, CD11b^−^CD27^+^, CD11b^+^ CD27^+^, and CD11b^+^ CD27^−^ [56]. By this criterion, it was shown that the majority of CD56^bright^ NK cells were immature CD11b^−^CD27^−^ and CD27^+^, whereas the majority of CD56^dim^ NK cells are mature CD11b^+^CD27^−^ [56]. Functionally, the CD11b^+^ CD27^−^ subset displayed the highest cytotoxicity, whereas the CD27^+^ subsets displayed high potential in secreting cytokines. The majority of NK cells in the peripheral blood were shown to be CD11b^+^ CD27^−^ (>95%) [56]. 

In healthy individuals, NK cells are widely distributed in the periphery. They not only circulate through the peripheral blood, but are also present in both lymphoid and non-lymphoid organs such as spleen, lymph nodes, liver, lung, and uterus. Contrary to their distribution in the peripheral blood, CD56^bright^CD16^−/lo^ NK cells are the major subset in many tissues, such as lymph nodes, tonsil, liver, and uterus [52]. It is noteworthy that the CD56^bright^ tissue-resident NK cell subset was shown to be phenotypically and functionally distinct from circulating CD56^bright^ NK cells [51,52,53]. Phenotypically, tissue-resident NK cells express unique sets of surface adhesion molecules, such as CD69, CCR5, CXCR6, CD103, and/or CD49a, to fulfill their retention in respective tissues [51]. For example, decidual NK (dNK), also known as uterine NK (uNK) cells, which accounts for ~70% lymphocyte during early pregnancy, are CD56^bright^, but uniquely express CD49a and CD9 on their surface [51]. Functionally, dNK cells serve as a critical regulator of pregnancy and are involved in trophoblast invasion and vascular remodeling by secreting various chemokines and pro-angiogenic factors [52,57]. The origin of tissue-resident NK cells is controversial. Some evidence suggests that pre-existing hematopoietic precursors in tissues can differentiate *in situ* into tissue-resident NK cells [58], while others argue that the circulating NK cells could adapt to the tissue microenvironment and acquire tissue-specific phenotype and functions [59].

## 4. Enrichment of NK Cell Subsets with Diminished Cytotoxicity in Tumors

Tumor-infiltrating NK cells often have diminished cytotoxic function with distinct phenotypes. Most studies reported that tumor infiltrating NKs are mostly CD56^bright^. Several studies have shown a significant enrichment of non-cytotoxic CD56^bright^perforin^low^ NK cell subsets in lung and breast tumors as compared with the matched normal tissues [60,61]. Consistently, other studies have shown that in breast cancer patients, the subset of poor cytotoxic CD56^bright^NKG2A^hi^CD16^low^KIR^lo^ NK cells was increased in tumor infiltrates and that the increase correlated with poor disease prognosis [62]. A significant enrichment of the poor cytotoxic CD56^bright^CD16^dim^ NK cell subset was also found in tumor infiltrates of melanoma and colon cancer [63].

The phenotype or function of NK cells in tumors is generally thought to be shaped by tumor microenvironment (TME) cues. There is evidence suggesting that chemokine milieu in the TME contributes to the accumulation of poor cytotoxic CD56^bright^ subset of NK cells [61]. In neoplastic lung and breast tissues, it was shown that chemokines, such as CXCL2, CX3CL1, CXCL2, CXCL1, and CXCL8, that are specifically attracting the CD56^dim^ NK cell subset, are downregulated, whereas chemokines more specific for attracting the CD56^bright^ NK cell subset, such as CCL5, CCL19, CXCL9, and CXCL10, are upregulated [61]. However, whether the accumulation of the poor cytotoxic NK cells in human tumors is because of TME-induced alterations in NK phenotype, preferential migration of NK cell subsets in response to specific chemokine cues in TME, or differential survival/proliferation ability of the NK subsets in TME, or potential trans-differentiation of NK cells, is not well defined. Nonetheless, these studies demonstrated the complexity of TME in skewing NK cell function.

## 5. Tumor-Associated Immature NK Cell Phenotype

NK cell function is associated with its maturation status. Tumor infiltrating NK cells often present an immature phenotype. In a B16F10 lung metastasis model, it was showed that impaired NK maturation in mice lacking neonatal Fc receptor associated with reduced tumor control [64]. In patients with hepatocellular carcinoma, accumulation of the immature CD11b^−^CD27^−^ NK cell subset in tumor infiltrates was shown to correlate with poor clinical outcome [65]. A substantial increase in the CD11b^−^CD27^−^ NK cells and a concomitant reduction in single and double positive NK populations were observed in the tumor tissues as compared with adjacent non-tumor and control liver tissues [65]. Moreover, the frequency of the CD11b^−^CD27^−^ NK cell subset correlated with the size of the resected tumors [65]. The CD11b^−^CD27^−^ NK population was shown to have impaired production of IFNγ, as well as poor cytotoxic potential [65].

Pre-clinical studies suggest tumor secreted soluble mediators can curtail NK cell maturation. Two studies from Richards and group have demonstrated defective NK maturation in the bone marrow of mice bearing tumors of breast, colon, melanoma, and lymphoma [66,67]. In the first study, they found a significant reduction in the mature CD11b^hi^ NK cells in the bone barrow of tumor-bearing mice as compared with non-tumor bearing control mice, suggesting an impact of tumor growth on the maturation status of NK cells [66]. A further study with adoptive transfer of bone-marrow derived immature CD11b^−^ NK cells into tumor bearing mice demonstrated that NK cell maturation was arrested at the CD11b^low^ stage [66]. In the second study, they showed that the tumor growth-associated reduction in NK cell numbers was attributed to the significant reduction in NK cell progenitors (CD122^+^NK1.1^−^DX5^−^CD3^−^) and common lymphoid progenitors (Lin^−^CD127^+^cKit^+^Sca^+^) with bone marrow transplant experiments [67]. Although underlying mechanisms associated with these observations were not fully dissected, the findings have evidently demonstrated that tumor-derived soluble factors negatively impact the lymphopoiesis and maturation process of NK cells. 

There is evidence that tumors can induce a reversal in the maturation status of NK cells. Using a transgenic spontaneous polyoma middle T antigen (pyMT) breast tumor mouse model, Krneta et al. demonstrated striking differences in maturity and activation markers in intra-tumoral NK cells versus splenic NK cells from the tumor bearing mice [68]. They demonstrated that NK cells from the tumors displayed highly immature CD27^low^CD11b^low^ phenotype and low expression of granzyme B and perforin when compared with the splenic NK cells. To further delineate that the tumor microenvironment can directly modulate NK cell maturation and function, they adoptively transferred labeled NKp46^+^ mature splenic NK cells from C57BL/6 mice into pyMT mice. Intriguingly, after only three days of adoptive transfer, the intra-tumoral NK cells displayed a significant reduction of maturity markers CD27 and DX5 (CD49b) as compared with splenic NK cells. Together, these studies suggest that the tumor microenvironment can re-direct NK cell maturation status and function. Evidently, it is important for further investigation to understand the underlying mediators and pathways.

## 6. Cancer-Induced Phenotypic Changes in NK

NK cell in cancer patients, whether from tumor infiltrates or peripheral blood, often have an altered surface receptor repertoire that undermines the capacity to recognize and control tumor cells. Downregulation of activating NK receptors (e.g., NCRs, NKG2D, and DNAM-1) represents one of the common mechanisms employed by a wide array of cancer types to evade NK-mediated immunity [69]. In hematopoietic cancers such as AML, patient-derived NK often exhibit downregulation of NCRs (NKp30, NKp44, NKp46), which is correlated with their reduced cytotoxicity ex vivo against autologous leukemic cells [70,71,72]. This phenotype is most likely AML-induced, as the restoration, complete or partial, in the expression level of these receptors was seen in patients with complete remission [71]. The observed reduction in cytotoxicity ex vivo was shown to be a result of impaired target cell recognition owing to downregulation of NCR, rather than any intrinsic defect in the cytolytic capacity of the AML patient derived NK cells [70]. Downregulation of NCR was also shown in patients with solid tumors, such as non-small-cell lung cancer, melanoma, and breast cancer [62,73,74,75,76,77]. In non-small-cell lung carcinoma (NSCLC), for instance, tumor infiltrating NK cells are associated with a distinct surface receptor profile, characterized by lower expression of NKp30, NKp80, NKG2D, DNAM-1, and CD16 [73]. Upregulation of the inhibitory receptor NKG2A is another modulation that impairs NK cell cytotoxicity in cancer patients. It was well demonstrated in patients with breast cancer and liver cancer that tumor infiltrating NK cells have upregulated NKG2A expression [62,78]. It is noteworthy that NCR downregulation not only limits NK cell recognition of tumor cells, but also potentially limits T-cell mediated anti-tumor responses. It has been shown that NK cells can eliminate immature dendritic cells (iDC) through activation of NKp30, which in turn favors the expansion and function of mature DCs [79,80]. Interruption of the interaction of NK with iDC could potentially lead to suboptimal DC maturation and priming of tumor antigen-specific T cells and otherwise iDC-induced tolerance against certain tumor antigen.

Beyond the modifications on NK functional receptors, tumor modification of NK homing or exhaustion status has also been reported. Comparative microarray of tumor infiltrating (TI)-NK versus circulating NK from the same NSCLC patients yielded a unique TI-NK signature that is enriched in genes associated with NK activation, cytotoxicity, and migration [74]. Specifically, the study showed overexpression of CXCR5 and CXCR6 and lower expression of CX3CR1 and S1PR1 (sphingosine 1-phosphate receptor 1, a migration-associated receptor) at the transcript level on the TI-NK as compared with the circulating NK cells [74]. A recent study in endometrial tumor demonstrated severe impairment in the numbers and cytotoxic function of tumor infiltrating NK cells compared with those in the adjacent non-tumor tissues [81]. The same group also identified CD103^+^ “tumor resident” and CD103^−^ “recruited” NK cell subset in cancer patients. As compared with the CD103^−^ NK cells, the CD103^+^ NK cells displayed a significant increase in the expression of co-inhibitory markers TIM3 and TIGIT, which have been associated with inept NK cell responses [81,82,83,84,85,86]. NK cells from the patients with lymph node metastatic diseases have upregulated expression of these co-inhibitory molecules as compared with those with only localized diseases, suggesting that tumor-induced modulations affect the phenotypic and functional features of NK cells with disease progression [87].

There is evidence for the presence of phenotypically and functionally unique NK subsets, different from the conventional CD56^bright^CD16^dim^ and CD56^dim^CD16^bright^ NK cell phenotype, in cancer patients. Mamessier et al. has described the CD56^bright^CD16^+^ and CD56^dim^CD16^−^ subsets in breast cancer patients and further demonstrated the enrichment of these subsets in relevance to disease progression [88]. They demonstrated that both the NK subsets, CD56^bright^CD16^+^ and CD56^dim^CD16^−^, had functional immature phenotype of CD27^+^CD117^+^ and were highly enriched in the peripheral blood of patients with advanced breast cancer as compared with subjects with benign tumor and localized diseases. Interestingly, they further found that the frequency of CD56^bright^CD16^+^ and CD56^dim^CD16^−^ subsets in mammary tumors reflected the frequency of respective subset population in the peripheral blood [88].

The collective evidence suggests that the impact of tumor microenvironment on NK cell function can be multifaceted. The underlying mechanism whereby tumor microenvironment skews NK cell phenotype and activity is not fully understood, although it has been proposed that various cancer-associated immunosuppressive factors are the key mediators. For instance, reactive oxygen species, TGF-β, PGE2, and IDO1, which are derived from tumor cells or other immunosuppressive cells in the TME, have been shown to contribute to downregulation of NK receptors, including the NCRs and NKG2D [75,89,90,91,92,93,94,95]. On the other hand, it has been shown in vitro that IL-15 can rescue or counteract tumor-induced downregulation of NK receptors [72,96], which supports the current in-development cytokine therapeutic approach in combination with NK cell-based therapy.

## 7. Tumor-Induced NK Functional Plasticity

Human peripheral blood NKs possess a certain degree of plasticity to acquire an altered functional phenotype [97,98]. In this section, we will present evidence of tumor-induced phenotypical and functional alterations in mature NK cells in mouse models and/or cancer patients (Table 1) and discuss potential mechanisms underlying these conversions. 

### 7.1. Tumor-Induced Conversion of NK to dNK-Like Cells 

A subset of NK cells displaying the “dNK-like” phenotype has been shown in the peripheral blood, pleural effusion, and TILs of patients with different types of cancer, including colorectal cancer and lung cancer [63,99,100,103,104]. Phenotypically, these NKs are predominantly CD56^bright^CD16^−/dim^ and express CD9 and/or CD49a. A number of studies have demonstrated that these “dNK-like” cells secrete pro-angiogenic factors such as VEGF and angiogenin [99,100]. It was shown that supernatants from cultured “dNK-like” cells promoted endothelial cell tube formation in vitro [100,104]. It was also shown that NKs isolated from colorectal cancer patients had upregulated invasion-associated enzymes related to the MMP9-TIMP2/9 axis, a feature similar to dNK in tissue remodeling and angiogenesis [100]. The presence of this population in cancer patients, but not healthy individuals, suggests that the dNK-like phenotype and function is driven by tumor-related factors. 

Among many tumor-secreted soluble mediators, TGF-β has been shown to mediate the conversion of classical NK cells into dNK-like cells. Treatment of NK cells from healthy donors with TGF-β1 can induce NK cells to produce elevated levels of VEGF and PIGF (placental growth factor, a member of the VEGF family) [99]. It was shown that upon exposure to TGF-β in vitro, CD16^+^ peripheral blood NKs could be converted to a CD16^−^ dNK-like phenotype with acquired expression of CD9 and CD49a on their surface [97]. Combining TGF-β with hypoxia and a demethylating agent (5-Aza) further skewed peripheral blood NK cells towards a dNK-like phenotype [98]. The converted CD16^−^CD9^+^CD49a^+^ NK cells displayed reduced cytotoxicity, secreted the pro-angiogenic factor VEGF, and were able to promote trophoblast tumor invasion [98]. With the evidence suggesting that circulating NKs could be recruited to the uterus during early pregnancy and undergo reprogramming in the uterine microenvironment [59], it is plausible to hypothesize that the tumor microenvironment with high levels of TGF-β and hypoxia can induce reprogramming of NK cells to dNK-like phenotype.

### 7.2. Tumor-Induced Conversion of NK Cell into Type I Innate Lymphoid Cells (ILC1)-Like Phenotype

NK cells and type I innate lymphoid cells (ILC1s) are two members of the group I innate lymphoid cells, grouped together based on their shared capacity to produce IFN-γ [105,106]. In contrast to circulating conventional NK cells, ILC1s consist of tissue-resident subsets and are rarely found in the circulation at steady state [107]. In mice, ILC1s are identified in tissues including spleen, liver, small intestine, salivary gland, and adipose tissues at steady state, and they are also found in tumor tissues [108,109,110,111,112,113,114]. In human, ILC1 subsets are described in tonsil, gut, and tumor tissues [105,115,116,117]. While conventional NK cells are considered to possess dual effector functions of both cytokine secretion and cytotoxicity, ILC1s are often viewed as potent cytokine secretors with poor cytolytic capacity [105,110]. Lineage tracing indicates that conventional NK cells and ILCs arise from distinct progenitors [110,111,118]. In mice, both transcriptional factors T-bet and Eomes are required for conventional NK cell development, whereas only T-bet is required for ILC1 development [119]. Human and mouse ILC1s have been associated with hallmarks of TGF-β imprinting, such as the expression of CD103 [112,115].

Mouse NK cells and ILC1s both express NK1.1 and NKp46 [105,106]. The two populations are often distinguished based on their differential expression of CD49a and Eomes. NK cells are CD49a^−^Eomes^+^, whereas ILC1s are CD49a^+^ Eomes^−^ [101]. At steady state, TRAIL, CD73, CD200R1, and CD61 could serve as additional surface markers for ILC1s [105,120]. The phenotypic distinction between NK cells and ILC1s in pathological conditions is not very well characterized. One study showed that CD49b^+^ Eomes^+^ mouse NK cells have increased expression of CD49a and CD61, with markers being associated with ILC1s in murine cytomegalovirus (MCMV)-infected mice [120]. Therefore, discrimination of NK cells and ILC1s solely based on surface markers is insufficient and needs to be combined with other criteria such as Eomes expression and expression of ILC1 signature genes [105,108]. 

It is well established that NK cells play a critical role in controlling tumor growth and metastasis [121,122,123,124,125]. It is unclear to what extent ILC1s participate in anti-tumor immunosurveillance, despite their phenotypical and functional similarities to NK cells. To date, there is little evidence to support inter-conversion between the NK cells and ILC1s under normal physiological conditions. However, the plasticity between mouse NK and ILC1 in the context of tumors was recently demonstrated [101]. Using a variety of mouse tumor models, Gao et al. showed that tumor-infiltrating NKs (CD49a^−^CD49b^+^) could be converted to an intermediate ILC1 (intILC1, CD49a^+^CD49b^+^) phenotype characterized by reduced Eomes and increased CD49a expression. Tumor-infiltrating intILC1s lost the ability to control tumor initiation and metastasis, and displayed profound phenotypical and transcriptomic alterations compared with TI-NKs [101]. It was shown that intILC1s are functionally impaired with upregulated expression of immune checkpoint receptors [101]. intILC1s was shown to be pro-tumorigenic. Mechanistically, it was shown that intILC1s secrete pro-angiogenic factors such as platelet-derived growth factor (PDGF)-AB and could promote endothelial tube formation in vitro [101]. In the same study, CXCR6^+^ ILC1-like cells were described in the peripheral and TILs of patients with gastrointestinal stromal tumors (GIST) [101], although whether these cells are converted from classical NK cells is unknown. 

Non-canonical TGF-β signaling has been implicated in promoting the conversion of conventional NK cells to an ILC1-like phenotype. Gao et al. demonstrated that the ILC1-like conversion of NK cells in the mouse models was dependent on non-canonical TGF-β signaling in NK cells, and that NK cells deficient in TGF-β signaling failed the conversion [101]. Cortez et al. further revealed that SMAD4, a central signal transducer of the canonical TGF-β signaling, could paradoxically act as a negative regulator of the conversion by curtailing non-canonical TGF-β signaling [102]. SMAD4-deficient mouse NK cells are hyper-responsive to TGF-β receptor 1 (TGFβ R1)-mediated signals. The SMAD4-deficeint NK cells isolated from the spleen, liver, and small intestine acquire ILC1 characteristic markers, such as CD49a, CD73, and TRAIL. These mice displayed impaired anti-tumor immunity [102]. A recent study by Rautela et al. showed that independent of TGF-β signaling, activin-A (a member of the TGF-β family) signaling could induce a similar ILC1-like conversion (increased CD49a and decreased Eomes expression) in conventional NK cells [126]. 

Recently, a new subset of regulatory CD3-CD56+ ILC subset in the TILs of patients with high-grade serous tumors has been described [127]. The presence of this subset TIL cultures correlates with a reduction in the time to disease progression. In vitro, this CD3^−^CD56^+^ subset was poorly cytotoxic, but suppressed TIL expansion and altered TIL cytokine production. These CD3^−^CD56^+^ cells presented a distinct cytokine profile from those of NK cells and other ILCs. It is unclear whether these cells were the result of conversion of NK cells in the tumor microenvironment, or whether they represent a novel, distinct ILC population. 

Beyond evidence demonstrating the tumor-induced conversion of NK cells to ILC1-like and dNK-like phenotypes, one study showed that NK cells could be converted to myeloid derived suppressor cells in tumor-bearing mice in a GM-CSF dependent manner [128]. However, to date, no other studies have reported similar results to support this observation. 

## 8. Concluding Remarks

Mature NK cells were considered terminally differentiated NK cells that play a significant role in controlling tumor growth and metastasis. Many NK cell-based therapies with the intention to harness NK cell receptors to enhance NK cell cytolytic function are currently being developed for treating solid tumors. Recent findings indicate that specific tumor environment cues can profoundly impact NK phenotype and function (Figure 1) and NK cell-based therapeutic efficacy, and more profoundly, could potentially convert therapeutic NK cells into tumor-promoting NK cells. Therefore, strategies to co-target tumor microenvironment, such as co-targeting hypoxia pathway, tumor released TGF-β, or non-canonical TGF-β signaling pathways, should be considered in order to achieve desired therapeutic efficacy. Evidently, further investigations to gain an in-depth understanding of the molecular mechanisms that drive NK cell functional plasticity are warranted. 

## Figures and Tables

**Figure 1 jcm-08-01492-f001:**
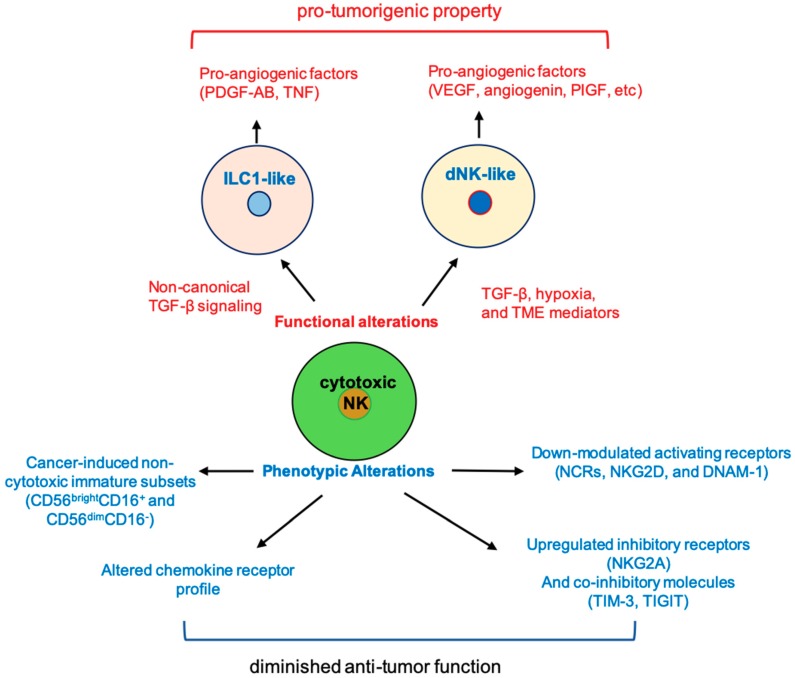
Cancer-induced phenotypic and functional alterations in natural killer (NK) cells. The function and phenotype of cytotoxic NK cells can be altered by tumor environment cues to become pro-tumorigenic or to lose anti-tumor function. PIGF, placental growth factor; ILC1, type I innate lymphoid cells; TME, tumor microenvironment; dNK, decidual NK; NCR, natural cytotoxicity receptor; DNAM, DNAX accessory molecule.

**Table 1 jcm-08-01492-t001:** Pro-tumor conversions of natural killers (NKs).

Type of Conversion	Mouse Models/Human Patients	Phenotype	Functions	Limitations
Conventional NK to decidual NK-like phenotype [99,100]	Human patients of non-small cell lung carcinoma and colorectal cancer	↑↑↑ ^1^ CD49a and/or CD9	↑↑↑ Secretion of pro-angiogenic factors and enzymes associated with tissue remodeling	The actual contribution to tumor progression in vivo is hard to determine
NK to ILC1-like phenotype [101,102]	Mouse models of fibrosarcoma, melanoma, and prostate cancer	↓↓↓ ^1^ Eomes and CD62L ↑↑↑ CD49a, CD69, DNAM-1 and TRAIL ↑↑↑ LAG-3, CD96, and CTLA-4	↑↑↑ Secretion of PDGF-AB and GM-CSF ↓↓↓ Secretion of RANTES	Human relevancy is unclear

^1^ ↑↑↑—upregulated; ↓↓↓—downregulated.

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
