# Peer review of "NK Cell Plasticity in Cancer"

_jcm, 2019, doi:10.3390/jcm8091492_

Round 1

Reviewer 1 Report

Dhar et al., summarize what it is known on the effect of cancer on NK cell plasticity and how they influence their functions, leading to tumor-promoting phenotypes. The authors are done a good job describing extensively the plasticity of NK cells. Unfortunately, there are not enough details on potential NK-cell based therapies, as mention in the abstract. I also feel the review will gain with a figure instead of a table to illustrate the plasticity of NK cells. I am also providing a list of suggestions and recommendations that the author may wish to follow. Also, some sentences need rewriting.  

L.25-26, 44-45, 67-70 and 90: missing words/simplify the sentences  

L.102: Typo CX3CL1

Reference to add part 4: Castaneda DC, Dhommée C, Baranek T, Dalloneau E, Lajoie L, Valayer A, Arnoult C, Demattéi M-V, Fouquenet D, Parent C, Heuzé-Vourc'h N and Gouilleux-Gruart V (2018) Lack of FcRn Impairs Natural Killer Cell Development and Functions in the Tumor Microenvironment. Front. Immunol. 9:2259. doi: 10.3389/fimmu.2018.02259

L.152: DNAM-1

L.158-159: Could you develop the idea of impaired target cell recognition? Are you talking about ligand downregulation? 

L.161: define NSCLC

L.176: define S1PR1

L.180-183: Observation of exhaustion via TIM3 and TIGIT has been demonstrated in T cells. It is still unclear for NK cells (Carotta S. Targeting NK cells for anticancer immunotherapy: clinical and preclinical approaches. Front Immunol. (2016) 7:152. doi: 10.3389/fimmu.2016.00152). Please rewrite. 

Table 1: The table is not easy to read and have several typos. I will suggest doing a figure comparing the plasticity of NK cells. 

L.209: Define CRC

L.215: Define PIGF

L.240: Define SMAD4

L.249: Typo "not unknown"

Conclusion: Instead of mentioning some therapies here I will suggest to talk about NK cell based-therapies in a separate part or maybe mentioning it at the end of the different parts. 

Author Response

We thank the reviewers for the recognition of the significance of our manuscript and the critical reading of the manuscript. We have now addressed all the comments point-by-point below. All the revisions are highlighted in yellow in the revised manuscript.

Reviewer 1.

Comments and Suggestions for Authors

Dhar et al., summarize what it is known on the effect of cancer on NK cell plasticity and how they influence their functions, leading to tumor-promoting phenotypes. The authors are done a good job describing extensively the plasticity of NK cells. Unfortunately, there are not enough details on potential NK-cell based therapies, as mention in the abstract. I also feel the review will gain with a figure instead of a table to illustrate the plasticity of NK cells. I am also providing a list of suggestions and recommendations that the author may wish to follow. Also, some sentences need rewriting.  

L.25-26, 44-45, 67-70 and 90: missing words/simplify the sentences  

Response: We have corrected and rephrased the sentences.

L.102: Typo CX3CL1

Response: We have corrected/re-written the sentences.

Reference to add part 4: Castaneda DC, Dhommée C, Baranek T, Dalloneau E, Lajoie L, Valayer A, Arnoult C, Demattéi M-V, Fouquenet D, Parent C, Heuzé-Vourc'h N and Gouilleux-Gruart V (2018) Lack of FcRn Impairs Natural Killer Cell Development and Functions in the Tumor Microenvironment. Front. Immunol. 9:2259. doi: 10.3389/fimmu.2018.02259

Response: We have added this reference.

L.152: DNAM-1

Response: We have corrected this typo throughout the text.

L.158-159: Could you develop the idea of impaired target cell recognition? Are you talking about ligand downregulation? 

Response: Here we intend to talk about impaired tumor cell recognition due to downregulation of NCRs on NK cells. We have revised the sentence to better convey the idea.

L.161: define NSCLC

Response: We have now defined the term.

L.176: define S1PR1

Response: We have now defined the term.

L.180-183: Observation of exhaustion via TIM3 and TIGIT has been demonstrated in T cells. It is still unclear for NK cells (Carotta S. Targeting NK cells for anticancer immunotherapy: clinical and preclinical approaches. Front Immunol. (2016) 7:152. doi: 10.3389/fimmu.2016.00152). Please rewrite. 

Response: We have modified the statement and included the relevant references.

Table 1: The table is not easy to read and have several typos. I will suggest doing a figure comparing the plasticity of NK cells. 

Response: We have modified the table and also included a figure.

L.209: Define CRC

Response: We have now defined the term.

L.215: Define PIGF

Response: We have now defined the term.

L.240: Define SMAD4

Response: We have now defined the term.

L.249: Typo "not unknown"

Response: We have corrected the typo.

Conclusion: Instead of mentioning some therapies here I will suggest to talk about NK cell based-therapies in a separate part or maybe mentioning it at the end of the different parts. 

Response: We have now included a section discussing NK cell based-therapies.

Reviewer 2 Report

Synopsis

The authors review Natural Killer (NK) cell phenotype and function as well as changes thereof in the context of tumour challenges. Different types of tumour-associated NK cells and their anti-tumour capacity and ability to modulate other anti-tumour responses are described. The current understanding of how the tumour microenvironment impacts/alters NK cells phenotypically is reviewed and an overview of how tumour-associated NK cells can convert to resemble tissue-resident populations of NK as well as type 1 innate lymphoid cells (ILC1) is provided. The authors surmise that an in depth understanding of phenotypic and functional changes in tumour-associated NK cells and their impact on overall tumour control is of paramount importance given the increasing interested in NK cell-based therapeutic approaches to cancer treatment.

Comments to authors

This is an interesting and highly topical review of how tumour microenvironment can shape NK cells and their anti-tumour responses. As the authors highlight in their manuscript, the rising interest in NK cells and their application in cell-based cancer treatments necessitates a fundamental understanding of how tissues/tumours impact NK cells and their ability to kill tumour cells or regulate immune responses. A solid synthesis of the most current literature on tumour-associated NK cell plasticity is provided. However, the review would benefit from a brief expansion on NK cell plasticity as well as the difference between NK cells and ILC1 in general. In this context, I believe it is important to mention that the field is currently cautious with the use of definitive expressions such transdifferentiation as the understanding of how NK cell phenotype and expression profile are impacted by soluble factors and tissue microenvironment is incomplete. With the above in mind, the following comments should be addressed in order to give a clearer and more comprehensive overview of the current understanding of how phenotypic changes in tumour-associated NK cells may impact tumour control.

Major Comments

Line 24 & 27 - The sentence on NK cell inhibitory receptor signalling is unclear and needs to be amended. The sentence starting in line 27 should also be corrected. Line 40 – The sentence referencing Curson et alis misleading as the gene signature analysis does not predict NK cell infiltration into the tumour bed per se but rather correlates a specific gene signature with increased NK cell infiltration. This should be phrased more clearly. Line 59 – Although Chiossone et alfurther investigate CD27/CD11b NK cell subsets the original paper on the differential functionality of CD27/CD11b NK cell populations is by Hayakawa and Smyth (CD27 Dissects Mature NK Cells into Two Subsets with Distinct Responsiveness and Migratory Capacity, J Immunology, 2006) and should also be referenced here. Line 78 – 79 – References 23 and 24 use the terminologies ‘NK cells’ and ‘trNK’ but are indeed mouse studies on liver ILC1s (CD49b- CD49a+ Eomes- TRAIL+ etc) and therefore not suitable to be referenced as tissue-resident NK cells. I believe there are more appropriate human studies on trNKs that are then also more comparable to the human study mentioned in Ref 25. Line – 106 - I caution from using the term ‘transdifferentiation’ as this delineates a reprogramming into a different somatic cell line and it is currently unknown whether NK cells truly undergo such fundamental changes. Indeed, recent studies avoid using this kind of definitive terminology and refer to NK cells with altered phenotype and expressional profile as ‘ILC1-like cells’ (i.e. Berrien-Elliott et al, MicroRNA-142 Is Critical for the Homeostasis and Function of Type 1 Innate Lymphoid Cells, Immunity, 2019). Line 119 – 135 – This paragraph provides an overly detailed outline of Ref 32 and 33 and should be shortened to summarize the relevant findings. Line 136 – as above for 5. Line 149 – The title of section 5 is ‘Cancer-induced phenotypic changes in NK’, however the previous section as also described phenotypic changes in NK cells (Ref 34). The manuscript should be reorganised accordingly. Line 196 – as for 5. above. Table 1 – The table requires a clearer structure as the current layout doesn’t allow an easy intake of the information provided. A suggestion might be using up and down arrows to indicate increase or decrease of expression as well as aligning the text to one side instead of in the centre. Furthermore, it would be more appropriate to list the conversion in the first row as NK to ILC1-like as Ref 72 uses this term in its title and the authors of Ref 64 now also reference this study as ILC1-like conversion of NK cells (Rautela et al, Therapeutic blockade of activin-A improves NK cell function and antitumour immunity, Science Signalling, 2019). Line 224 – please change trans-differentiate as for 5. Line 229 – Although NK cells and ILC1 differentially express CD49a and CD49b in some tissues at steady state, these markers are not sufficient to fully distinguish NK cell and ILC1 populations. Additional markers such as CD200r1, CD61 as well as the expression of the transcription factor Eomesodermin are required (Weizmann et al, ILC1 confer early host protection at initial sites of viral infection, Cell 2017 and reviewed in Colonna, Innate Lymphoid Cells: Diversity, Plasticity and Unique Function, Immunity, 2018). Also, a further brief discussion of the different properties of NK cells and ILC1s would add value to the review. Line 240 – Ref 72 Cortez et al may not investigate NK cells in the context of anti-tumour responses but still provides important insights on the impact of TGF-b on NK cells and as such sheds more light on the TGF-b-mediated changes observed in the tumour models studied in Ref 64. This should be discussed in greater detail. Line 267 – as for 5.

Minor Comments

Line 30 – 32, NK cells can also impact immune responses through the production of factors such as IL-10 and GM-CSF. Line 40 – It would be appropriate reference 4. Bottcheret al Line 44 – The sentence is unclear and requires correction. Line 97 – I believe this sentence should read ‘…. CD56bright CD16+ and CD56dimCD16- subsets in….’ Line 146 – Sentence is unclear and needs to be corrected. Line 161 - Define the acronym NSCLC Line 258 – Define acronym MDSC

Author Response

We thank the reviewers for the recognition of the significance of our manuscript and the critical reading of the manuscript. We have now addressed all the comments point-by-point below. All the revisions are highlighted in yellow in the revised manuscript.

Reviewer 2

Comments and Suggestions for Authors

Synopsis

The authors review Natural Killer (NK) cell phenotype and function as well as changes thereof in the context of tumour challenges. Different types of tumour-associated NK cells and their anti-tumour capacity and ability to modulate other anti-tumour responses are described. The current understanding of how the tumour microenvironment impacts/alters NK cells phenotypically is reviewed and an overview of how tumour-associated NK cells can convert to resemble tissue-resident populations of NK as well as type 1 innate lymphoid cells (ILC1) is provided. The authors surmise that an in depth understanding of phenotypic and functional changes in tumour-associated NK cells and their impact on overall tumour control is of paramount importance given the increasing interested in NK cell-based therapeutic approaches to cancer treatment.

Comments to authors

This is an interesting and highly topical review of how tumour microenvironment can shape NK cells and their anti-tumour responses. As the authors highlight in their manuscript, the rising interest in NK cells and their application in cell-based cancer treatments necessitates a fundamental understanding of how tissues/tumours impact NK cells and their ability to kill tumour cells or regulate immune responses. A solid synthesis of the most current literature on tumour-associated NK cell plasticity is provided. However, the review would benefit from a brief expansion on NK cell plasticity as well as the difference between NK cells and ILC1 in general. In this context, I believe it is important to mention that the field is currently cautious with the use of definitive expressions such transdifferentiation as the understanding of how NK cell phenotype and expression profile are impacted by soluble factors and tissue microenvironment is incomplete. With the above in mind, the following comments should be addressed in order to give a clearer and more comprehensive overview of the current understanding of how phenotypic changes in tumour-associated NK cells may impact tumour control.

Major Comments

Line 24 & 27 - The sentence on NK cell inhibitory receptor signalling is unclear and needs to be amended. The sentence starting in line 27 should also be corrected.

Response: We have now re-written the section.

Line 40 – The sentence referencing Curson et alis misleading as the gene signature analysis does not predict NK cell infiltration into the tumour bed per se but rather correlates a specific gene signature with increased NK cell infiltration. This should be phrased more clearly.

Response: We have re-phrased the sentence.

Line 59 – Although Chiossone et alfurther investigate CD27/CD11b NK cell subsets the original paper on the differential functionality of CD27/CD11b NK cell populations is by Hayakawa and Smyth (CD27 Dissects Mature NK Cells into Two Subsets with Distinct Responsiveness and Migratory Capacity, J Immunology, 2006) and should also be referenced here.

Response: We have now added the reference.

Line 78 – 79 – References 23 and 24 use the terminologies ‘NK cells’ and ‘trNK’ but are indeed mouse studies on liver ILC1s (CD49b- CD49a+ Eomes- TRAIL+ etc) and therefore not suitable to be referenced as tissue-resident NK cells. I believe there are more appropriate human studies on trNKs that are then also more comparable to the human study mentioned in Ref 25.

Response: We thank the reviewer for pointing out this mistake. We have re-written the part and replaced the studies on liver ILC1s with a more relevant reference.

Line – 106 – I caution from using the term ‘transdifferentiation’ as this delineates a reprogramming into a different somatic cell line and it is currently unknown whether NK cells truly undergo such fundamental changes. Indeed, recent studies avoid using this kind of definitive terminology and refer to NK cells with altered phenotype and expressional profile as ‘ILC1-like cells’ (i.e. Berrien-Elliott et alMicroRNA-142 Is Critical for the Homeostasis and Function of Type 1 Innate Lymphoid Cells, Immunity, 2019).

Response: In the revised manuscript, we have replaced the term “transdifferentiation” with more appropriate phrases.

Line 119 – 135 – This paragraph provides an overly detailed outline of Ref 32 and 33 and should be shortened to summarize the relevant findings.

Response: We have now removed the details and revised the section as a summary of key findings of the studies.

Line 136 – as above for 5. Line 149 – The title of section 5 is ‘Cancer-induced phenotypic changes in NK’, however the previous section as also described phenotypic changes in NK cells (Ref 34). The manuscript should be reorganised accordingly.

Response: We have re-organized the sections in the manuscript. The study mentioned is now in the section “cancer-induced phenotypic changes in NK cells”.

Line 196 – as for 5. above.

Response: We have changed the term “trans-differentiation”.

Table 1 – The table requires a clearer structure as the current layout doesn’t allow an easy intake of the information provided. A suggestion might be using up and down arrows to indicate increase or decrease of expression as well as aligning the text to one side instead of in the centre. Furthermore, it would be more appropriate to list the conversion in the first row as NK to ILC1-like as Ref 72 uses this term in its title and the authors of Ref 64 now also reference this study as ILC1-like conversion of NK cells (Rautela et al, Therapeutic blockade of activin-A improves NK cell function and antitumour immunity, Science Signalling, 2019).

Response: We have modified the table and also included a figure now.

Line 224 – please change trans-differentiate as for 5.

Response: We have changed the term “trans-differentiation”.

Line 229 – Although NK cells and ILC1 differentially express CD49a and CD49b in some tissues at steady state, these markers are not sufficient to fully distinguish NK cell and ILC1 populations. Additional markers such as CD200r1, CD61 as well as the expression of the transcription factor Eomesodermin are required (Weizmann et al, ILC1 confer early host protection at initial sites of viral infection, Cell 2017 and reviewed in Colonna, Innate Lymphoid Cells: Diversity, Plasticity and Unique Function, Immunity, 2018). Also, a further brief discussion of the different properties of NK cells and ILC1s would add value to the review.

Response: We have revised the section per the reviewer’s recommendation. We have now included a brief discussion about different properties of NK cells and ILC1s.

Line 240 – Ref 72 Cortez et al may not investigate NK cells in the context of anti-tumour responses but still provides important insights on the impact of TGF-b on NK cells and as such sheds more light on the TGF-b-mediated changes observed in the tumour models studied in Ref 64. This should be discussed in greater detail.

Response: We have now discussed the study by Cortez et al. in more detail.

Line 267 – as for 5.

Response: We have changed the term “trans-differentiation”.

Minor Comments

Line 30 – 32, NK cells can also impact immune responses through the production of factors such as IL-10 and GM-CSF. Line 40 – It would be appropriate reference 4. Bottcheret al 

Response: We have revised according to the reviewer’s suggestions. 

Line 44 – The sentence is unclear and requires correction.

Response: we have corrected the sentence, now as in line 65-66 in the revision.

Line 97 – I believe this sentence should read ‘…. CD56bright CD16+ and CD56dimCD16- subsets in….’

Response: We have made correction, now as in line 297 in the revision.

Line 146 – Sentence is unclear and needs to be corrected.

Response: We have corrected and rephrased the sentence, as in line 242-244 in the revision.

Line 161 - Define the acronym NSCLC Line 258 – Define acronym MDSC 

Response: MDSC is defined, as in line 404-405 in the revision.

Reviewer 3 Report

In this manuscript Payal Dhar et al. consider the Natural Killer cells critical immune components in controlling tumor growth and dissemination and the potential functional plasticity of NK cells in the context of tumors.

Comments:

1) the review is well written and easy to understand. Very useful is the table 1 of pro-tumor conversion of NK cells.

2) The authors should address these major concerns:

   - in the article there is no mention about clinical NK92 cell line and clinical   trials using allogenic NK 92 cells as therapies; 

   - the authors should discuss also NK cells for cancer immunothrepy using CAR. (Cell Stem Cell of June 28, 2018 from Dan Kaufman's group).

  - NK cells counts are associated with Molecular Relapse-Free survival after Imatinib discontinuation in Chronic Myeloid Leukemia (CML). The authors should mention the IMMONOSTIM study and the role of NK cells in CML together with AML.

3) The authors should check all the references and put more references updated in the window 2015-2019. On a total of 74, less than 20 are in that window.

Author Response

We thank the reviewers for the recognition of the significance of our manuscript and the critical reading of the manuscript. We have now addressed all the comments point-by-point below. All the revisions are highlighted in yellow in the revised manuscript.

Reviewer 3.

Comments:

1) the review is well written and easy to understand. Very useful is the table 1 of pro-tumor conversion of NK cells.

2) The authors should address these major concerns:

in the article there is no mention about clinical NK92 cell line and clinical   trials using allogenic NK 92 cells as therapies; 

Response: We have now included a section discussing various NK-based therapies, and the clinical trials.

the authors should discuss also NK cells for cancer immunothrepy using CAR. (Cell Stem Cellof June 28, 2018 from Dan Kaufman's group).

Response: We have now discussed the study in the section about NK-based therapies.

NK cells counts are associated with Molecular Relapse-Free survival after Imatinib discontinuation in Chronic Myeloid Leukemia (CML). The authors should mention the IMMONOSTIM study and the role of NK cells in CML together with AML.

Response: We have now discussed the IMMUNOSTIM study in the introduction section.

3) The authors should check all the references and put more references updated in the window 2015-2019. On a total of 74, less than 20 are in that window.

Response: We have added more up-to-date references in the revised manuscript.

Round 2

Reviewer 2 Report

All my concerns have been addressed.

Reviewer 3 Report

The editing and the parts added are now in line and the manuscript is suitable for publication.

Sincerely